# A foundation model for human-AI collaboration in medical literature mining

Zifeng Wang[1] ✉, Lang Cao[2], Qiao Jin [3], Joey Chan[3], Nicholas Wan[3], Behdad Afzali [4], Hyun-Jin Cho[5], Chang-In Choi [5], Mehdi Emamverdi[6], Manjot K. Gill [7], Sun-Hyung Kim[5,8], Yijia Li [9], Yi Liu[10], Yiming Luo[11], Hanley Ong[12], Justin F. Rousseau [13,14], Irfan Sheikh[13], Jenny J. Wei[15], Ziyang Xu[16], Christopher M. Zallek[17], Kyungsang Kim[5], Yifan Peng [12,18,19], Zhiyong Lu[3] & Jimeng Sun [1,2,20] ✉

Applying artificial intelligence (AI) for systematic literature review holds great potential for enhancing evidence-based medicine, yet has been limited by insufficient training and evaluation. Here, we present LEADS, an AI foundation model trained on 633,759 samples curated from 21,335 systematic reviews, 453,625 clinical trial publications, and 27,015 clinical trial registries. In experiments, LEADS demonstrates consistent improvements over four cutting-edge large language models (LLMs) on six literature mining tasks, e.g., study search, screening, and data extraction. We conduct a user study with 16 clinicians and researchers from 14 institutions to assess the utility of LEADS integrated into the expert workflow. In study selection, experts using LEADS achieve 0.81 recall vs. 0.78 without, saving 20.8% time. For data extraction, accuracy reached 0.85 vs. 0.80, with 26.9% time savings. These findings encourage future work on leveraging high-quality domain data to build specialized LLMs that outperform generic models and enhance expert productivity in literature mining.

Literature mining, such as systematic review and meta-analysis, has become increasingly critical in medicine, serving as a vital route for discovering, integrating, and interpreting emerging research[1,2]. The proliferation of systematic reviews exemplifies the growing importance of literature mining, with over 50,000 review articles published annually on PubMed[3,4]. However, the process is costly and time-consuming. A study of 195 reviews indicated an average completion time of 67.3 weeks for systematic reviews[5]. Furthermore, a

[1]Keiji AI, Seattle, WA, USA. [2]School of Computing and Data Science, University of Illinois Urbana-Champaign, Urbana, IL, USA. [3]Division of Intramural Research, National Library of Medicine, National Institutes of Health, Bethesda, MD, USA. [4]Kidney Diseases Branch, National Institute of Diabetes and Digestive and Kidney Diseases, National Institutes of Health, Bethesda, MD, USA. [5]Center for Advanced Medical Computing and Analysis, Department of Radiology, Massachusetts General Hospital and Harvard Medical School, Boston, MA, USA. [6]National Eye Institute, National Institutes of Health, Bethesda, MD, USA. [7]Department of Ophthalmology, Northwestern University Feinberg School of Medicine, Chicago, IL, USA. [8]Division of Pulmonary and Critical Care Medicine, Department of Medicine, Chungbuk National University Hospital, Chungbuk National University College of Medicine, Cheongju, Republic of Korea. [9]Department of Medicine, University of Pittsburgh Medical Center, Pittsburgh, PA, USA. [10]Department of Medicine, Weill Cornell Medicine, New York, NY, USA. [11]Division of Rheumatology, Department of Medicine, Columbia University Irving Medical Center, New York, NY, USA. [12]Department of Radiology, Weill Cornell Medicine, New York, NY, USA. [13]Department of Neurology, UT Southwestern Medical Center, Dallas, TX, USA. [14]Clinical Informatics Center, University of Texas Southwestern Medical Center, Dallas, USA. [15]Department of Dermatology, University of Washington, Seattle, WA, USA. [16]Department of Dermatology, NYU Langone Health, New York, NY, USA. [17]OSF HealthCare Illinois Neurological Institute, Peoria, IL, USA. [18]Department of Population Health Sciences, Weill Cornell Medicine, New York, NY, USA. [19]Institute of Artificial Intelligence for Digital Health, Weill Cornell Medicine, New York, USA. [20]Carle Illinois College of Medicine, University of Illinois Urbana-Champaign, Urbana, IL, USA. ✉e-mail: zifeng@keiji.ai; jimeng@illinois.edu

study on the top NIH-funded institutions and pharmaceutical companies reports that each organization incurs costs of approximately $17 million annually to perform systematic literature reviews[6]. The challenge is further compounded by the sheer volume of medical literature, with PubMed now indexing over 35 million publications and receiving more than 1 million new entries each year[4]. A fair amount of them have poor metadata indexed and caused suboptimal study search precision[7]. As a result, researchers face mounting obstacles in conducting comprehensive literature mining, as evidenced by a review of caveats in 485 systematic reviews, including insufficient literature searches, potential study selection bias, and data extraction errors[8]. Beyond the meta-analysis use case, applications of literature mining also comprise the creation of new evidence[9], the revision of clinical guidelines[10], and the acceleration of drug discovery and development[11].

Recent advances in artificial intelligence (AI) have shown promise in transforming medical literature mining[12,13]. For instance, AI has been adopted for keyword generation to enhance literature search[14,15], facilitate study screening through retrieval[16,17], and support key entity extraction, including the identification of population, intervention, comparison, and outcomes (PICO) elements[18,19]. AI was also employed to summarize evidence from scientific publications[20–22]. The most recent developments in this domain are primarily driven by AI foundation models, particularly large language models (LLMs) like ChatGPT[23], which serve as generalist AI capable of adapting to diverse tasks[24]. These foundation models are typically adapted to medical tasks through two primary methods[25]: prompting, such as in-context learning (ICL)[26], chain-of-thought[27], and retrieval-augmented generation (RAG)[28]; and fine-tuning for specific tasks, such as named entity recognition[29] and evidence summarization[30].

Despite these advancements, several critical challenges persist. First, existing medical AI models are predominantly task-specific and narrow in scope, typically developed and tested on limited datasets[31]. These models often require fixed-format inputs and necessitate retraining for new tasks or varying data formats, failing to function as generalist AI capable of handling flexible inputs and generalizing across diverse topics[24]. Second, in our previous research, we developed TrialMind that prompts general-domain LLMs for multiple literature mining tasks[32]. Based on TrialMind, we have developed a platform to enable human-AI interaction[33]. Nonetheless, these adaptations may fall short of the effectiveness demonstrated by domain-specific fine-tuned models, as verified by the recent advancements in developing specialized medical LLMs[34–36]. Third, there is a limitation in comprehensively assessing AI methods' performance in literature mining tasks. Existing research has been constrained by limited sample size, with studies typically conducted on a scale of tens of systematic reviews and often focused exclusively on single tasks, e.g., search query generation[14,15] and citation screening[37–39]. This narrow scope may not adequately represent the full complexity of medical literature mining. Lastly, current validation efforts have primarily focused on AI's potential to automate processes, where critical challenges such as hallucination may happen and are not sufficiently addressed[40]. Given the high standards for accuracy and factual integrity in literature mining, developing and evaluating AI through human-AI collaboration presents a more pragmatic and reliable approach[41,42].

In this study, we introduce a foundation Large language model to facilitate human-AI collaboration in sEAisrch, screening, and Data extraction from medical literature Studies (LEADS). Our approach decomposes literature mining into subtasks, including search query generation, study eligibility assessment, study characteristics extraction, participant statistics extraction, arm design extraction, and trial result extraction (Fig. 1a). LEADS is constructed on a generic LLM and then fine-tuned using LEADSInstruct, an expansive instruction dataset curated from 21,335 systematic reviews involving 453,625 publications including 8485 systematic reviews with 27,015 clinical trial registries.

This comprehensive training strategy enables LEADS to achieve multitasking capabilities, handle flexible input requests, and generalize across diverse literature topics without requiring additional fine-tuning. In our experiments on broad review topics with thousands of systematic reviews, LEADS yields all-around superiority over cutting-edge generic LLMs like GPT-4o across all target tasks (Fig. 1d). This is further validated by a pseudo-prospective evaluation involving the reviews published after 2025 in the test, where LEADS shows comparable performance to GPT-4o and Deep Research. To validate the model's practical utility, we conducted a user study involving fourteen clinicians and two medical researchers across fourteen different institutions. The study compared two experimental arms: an Expert-only approach and an Expert+AI collaborative approach. Our findings reveal that LEADS (i.e., Expert+AI arm) provides encouraging benefits in accelerating citation screening and data extraction tasks while maintaining or surpassing the performance of manual efforts.

## Results
### Overview of LEADS and LEADSInstruct
LEADS addresses three fundamental tasks in systematic review methodology[43]: literature search, citation screening, and data extraction. To optimize the literature mining process, we decomposed these tasks into six specialized subtasks: (1) search query generation to maximize study identification coverage; (2) study eligibility assessment to systematically evaluate candidate citations; and (3–6) four distinct extraction subtasks: study characteristics, arm design, participant statistics, and results. Each subtask was formulated as a paired input-output instruction format suitable for Large Language Model (LLM) processing (see details in "Methods" section).

Our dataset comprises 21,335 systematic reviews from PubMed with their associated 453,625 publication citations, including 8485 reviews linked to 27,015 clinical trial records in ClinicalTrials.gov (Fig. 1a). We also built an instruction dataset, named LEADSInstruct, leveraging the linkage between systematic reviews, publications, and clinical trials (see details in "Methods" section). LEADSInstruct comprises 633,759 instruction data points across six tasks. The distributions of the most frequent conditions and interventions are illustrated in Fig. 1b, c. We fine-tuned a pre-trained Mistral-7B model[44] on LEADSInstruct using instruction tuning. For comparison, we also evaluated proprietary LLMs, including GPT-4o[45], GPT-3.5[46], and Haiku-3[47], open-source generic LLMs like Mistral[44] and Llama[48], and specialized medical LLMs such as BioMistral[49] and MedAlpaca[50]. We sampled 20% data to build the testing sets, which include thousands of systematic reviews and hundreds of thousands of clinical studies. To eliminate the data leakage risk, i.e., the published reviews and papers are in the LLMs' pretraining data, we also created a pseudo-prospective evaluation set, consisting of 31 reviews published after 2025. To our knowledge, LEADSInstruct constitutes the largest benchmark dataset to date for assessing AI performance in literature mining tasks.

### Synthesizing literature search queries for target studies
We evaluated LEADS's performance on publication and clinical trial search tasks. The system takes a research question as input and generates optimized search terms, which are then used to query PubMed or ClinicalTrials.gov for relevant publications or trial records (Fig. 2a). Our test set encompasses over 10,000 systematic reviews across diverse therapeutic areas (Fig. 2b). For each review, we calculated the Recall metric, measuring the proportion of relevant studies successfully retrieved by the search strategy. To establish comprehensive benchmarks, we implemented four distinct approaches with baseline LLMs: (1) zero-shot querying, where models generated search terms directly from the research question without examples; (2) few-shot prompting, which provided example search queries as guidance; (3) in-context learning (ICL), incorporating detailed expert-like guidance for query formulation; and (4) a hybrid

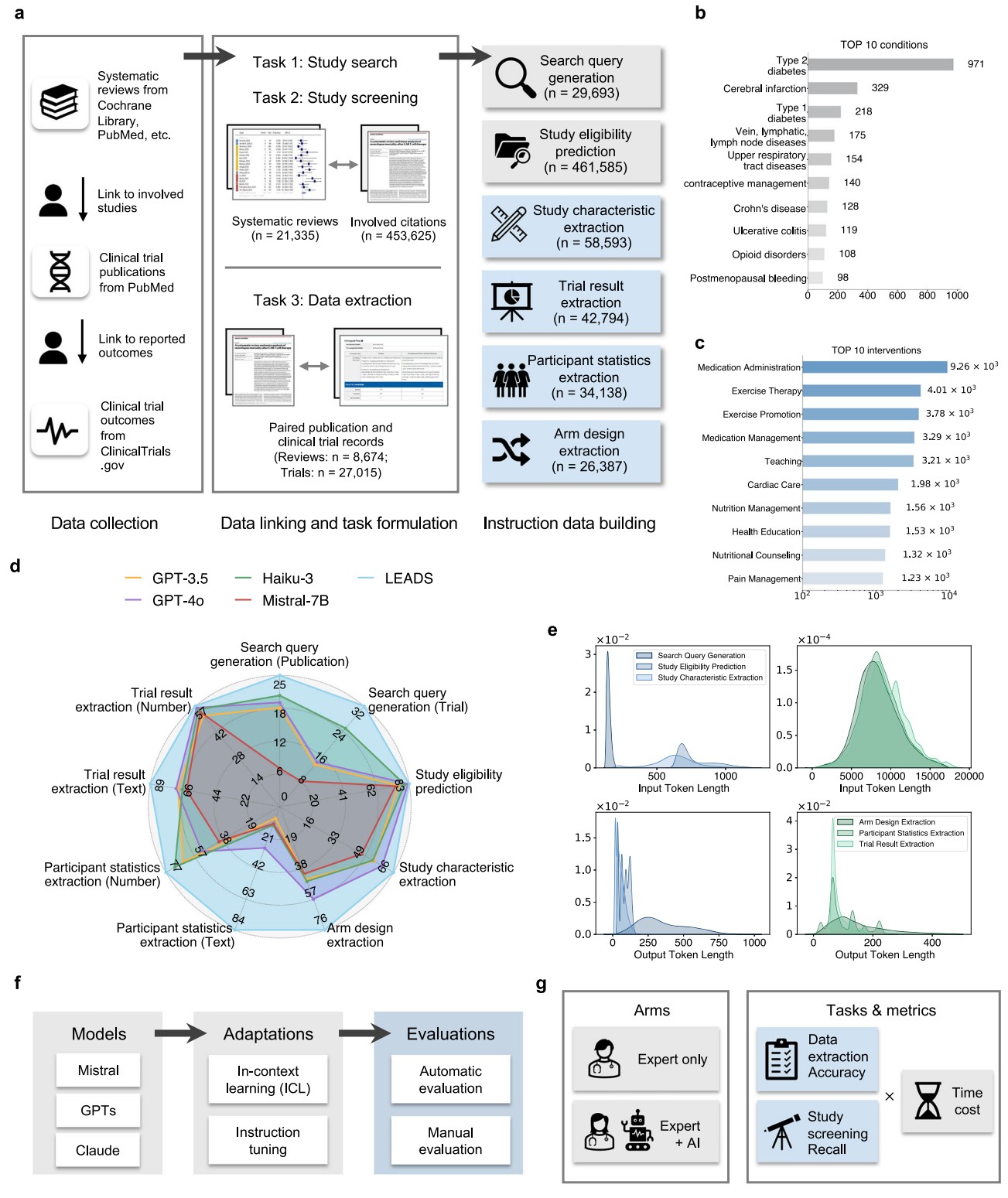

**Fig. 1 | Overview of LEADS and LEADSInstruct. a** LEADSInstruct consists of 20K+ systematic reviews, 453K+ publications, and 27K+ clinical trials linked across data sources. A hybrid approach is adopted to transform the linked data into instruction data covering six tasks in literature mining. **b** Bar plot showing the number of reviews covering different conditions. **c** Bar plot showing the number of reviews covering different interventions. **d** Comparative performance analysis contrasting LEADS with cutting-edge proprietary AI and open-source AI models. The evaluation metrics include Recall for search query generation, Recall@50 for study eligibility assessment, and Accuracy for the remaining tasks. **e** Density plot of the number of tokens in the inputs and outputs of the instruction datasets. **f** Illustration of experimental setups. **g** Illustration of the user study setup.

approach combining ICL with few-shot strategies to maximize performance (Extended Fig. 18).

The overall Recall is summarized in Fig. 2c. LEADS achieved Recall scores of 24.68 and 32.11 for the two tasks, surpassing the best-performing baselines by 3.76 and 7.43, respectively. A similar conclusion can be made in the pseudo-prospective evaluation of study search performance, where LEADS gets 0.30 Recall improvement over GPT-4o and Deep Research (Extended Fig. 1). Notably, LEADS, fine-tuned on

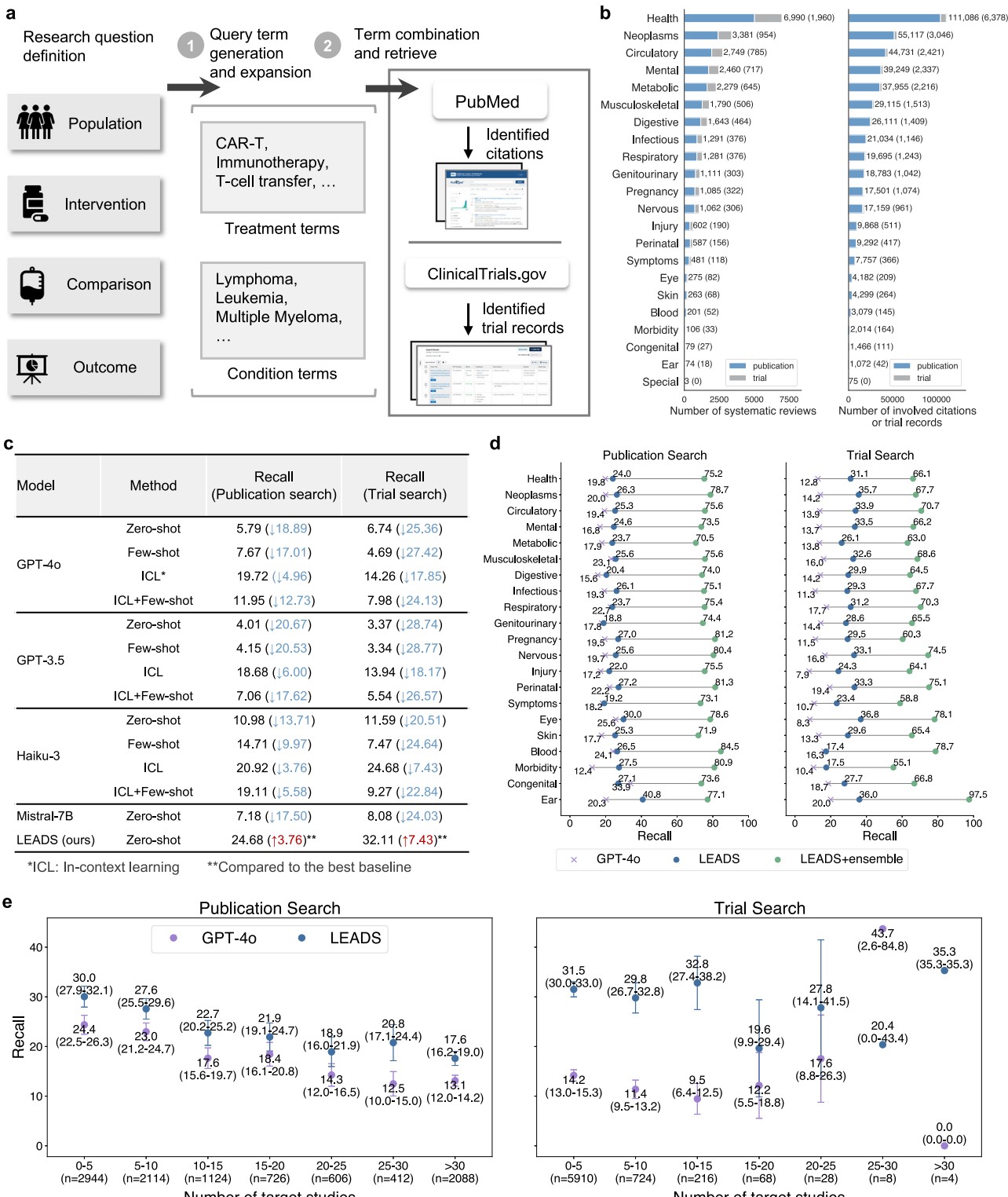

**Fig. 2 | LEADS performs literature search tasks. a** Illustration of how LEADS receives the research question definition, performs search query generation, and retrieves citations from the literature. **b** Distribution of the condition topics of the reviews and involved citations in the dataset. **c** Search query generation performance of LEADS and the leading models, in terms of the Recall achieved by the identified studies. The information in parentheses indicates the performance change of baselines compared to LEADS or LEADS compared to the best baseline in the same task. **d** Topic-wise comparison of LEADS to GPT-4o in terms of the Recall yielded by the generated search query. LEADS+ ensemble indicates an ensembling of multiple search queries. **e** Performance of LEADS and GPT-4o regarding the varied number of target studies for each review. The error bar indicates a 95% confidence interval of mean values, omitted when the sample size is smaller than ten.

Mistral-7B, demonstrated a significant improvement over the original Mistral model, which only achieved Recall scores of 7.18 and 8.08. This indicates a substantial improvement of 17.5 and 24.03, respectively, achieved by fine-tuning a generic LLM (Mistral-7B) on LEADSInstruct. Similarly, zero-shot generalist LLMs also performed notably worse, where GPT-4o yielded Recall scores of only 5.79 and 6.74 for publication and trial search tasks, respectively. This underscores the limitations of generic LLMs in handling domain-specific tasks without adaptation. Interestingly, adding examples to the prompts offered little to no benefit for most cases. For example, the ICL+Few-shot strategy with GPT-4o achieved a Recall of 11.95 for publication search, which was lower than the ICL strategy alone. This suggests that the diversity of review topics poses a challenge, as a few examples are insufficient to generalize across the wide range of therapeutic areas. In our evaluation, Recall is calculated as the recall at $K$, where $K$ represents the number of target studies in the original review. Although increasing $K$ will lead to higher recall, this strict metric remains valid for assessing our methods' performance. It is important to note that the original reviews themselves are not exhaustive; therefore, many relevant and newly identified studies retrieved by LEADS may not be included in the original citation list.

Figure 2d presents a topic-wise breakdown of Recall. Across all review topics, LEADS consistently outperformed GPT-4o, with Recall margins exceeding 5 in most cases. Similarly, for the trial search task, LEADS achieved Recall scores nearly double those of GPT-4o in most areas. These results highlight the effectiveness of the proposed instruction data generation pipeline, enabling LEADS to learn from the optimized synthetic query terms and outperform GPT-4o. Notably, the Recall reported for LEADS and baselines is based on a single pass for a fair comparison. In practical applications, however, an ensemble approach can be employed, where multiple sets of search terms are generated by LEADS running ten times, and the aggregated results are used to maximize coverage. We refer to this strategy as LEADS + Ensemble. This approach significantly improves performance, achieving a three- to four-fold increase in Recall compared to the single-pass LEADS, with average Recall scores exceeding 70 for publication search and 65 for trial search tasks.

We further examined how the difficulty of the search task affects the performance (Fig. 2e). Reviews were grouped based on the number of ground-truth studies to be identified in the search process. The more ground-truth studies, the more challenging the search is to identify all of them when only considering a fixed number of top-$K$ search results. Both methods showed a clear decreasing trend in Recall with increasing difficulty. For example, for reviews with 0–5 target studies, LEADS achieved a Recall of 30.0 compared to GPT-4o's 24.4. For reviews with 15–20 target studies, LEADS maintained a Recall of 21.9, outperforming GPT-4o's 18.4. Despite that, LEADS consistently outperformed GPT-4o across all bins. In the trial search task, this trend was less pronounced. LEADS achieved a Recall exceeding 25 across most bins, whereas GPT-4o consistently performed lower, with a Recall of around 10.

### Automated assessment and ranking of study eligibility

After identifying citations during the study search stage, the next step is to determine each citation's eligibility based on the predefined inclusion and exclusion criteria (Fig. 3d). LEADS uses the PICO elements defined in the target review to make criterion-level predictions for each citation, classifying them as Yes, Partially Yes, No, or Uncertain. We aggregate the criterion-level assessments into an overall eligibility score to rank the citations ("Methods" section). We evaluated LEADS using a dataset of 200 randomly sampled systematic reviews, each associated with 2000 candidate citations to be screened. This resulted in a total test size of 400,000 review-to-citation pairs. We compared LEADS against GPT-4o, GPT-3.5, Haiku, and Mistral-7B, and a

vector-based similarity ranking approach using OpenAI embeddings (referred to as the Dense method)[51]. Figure 3a illustrates the Recall@50 performance, where LEADS achieves performance comparable to GPT-4o, outperforming it in seven out of ten topics, despite being a much smaller model. Additionally, LEADS consistently achieves Recall scores above 80. In the pseudo-prospective evaluation, LEADS also shows comparable performance to GPT-4o, obtaining a Recall@50 of 85 versus GPT-4o's 86 (Extended Fig. 2).

Figure 3b presents the performance with varying shortlist lengths, measured by Recall@K, with $K$ ranging from 10 to 100. As $K$ increases, the difficulty of identifying all target studies decreases. The results show that LLMs generally outperform the Dense method, as they leverage natural language understanding to interpret the criteria text and comprehend the content of citations. The open-source Mistral model, representing LEADS before instruction tuning, performs significantly worse than proprietary LLMs. However, LEADS, after being fine-tuned with domain-specific instruction data, outperforms most proprietary LLMs and delivers performance comparable to GPT-4o.

Figure 3c compares the performance of LEADS, Mistral, and the Dense method across review groups with varying numbers of target studies. Generally, as the number of target studies increases, the task becomes more challenging, requiring a higher proportion of target studies to appear in the top-$K$ results. This increased difficulty is reflected in the decreasing trend observed for the two baseline methods. Mistral performs comparably to the Dense method when the number of target studies is fewer than 15. In contrast, LEADS maintains robust performance, showing no significant decline until the number of target studies exceeds 25. For example, in the "0–5" target studies group, the Recall scores are 0.81 for Dense, 0.84 for Mistral, and 0.90 for LEADS. In the "20–25" group, the scores are 0.70 for Dense, 0.76 for Mistral, and 0.87 for LEADS.

### Streamlined data extraction from scientific papers

LEADS follows the defined data fields and extracts the data from clinical research papers (Fig. 4d). A series of example inputs and outputs for these data extraction tasks can be found in Extended Figs. 8, 9, 10, and 11. The automatic evaluation results are shown in Fig. 4a. For numeric fields, exact match accuracy was used as the metric. For text fields, correctness was determined based on a similarity threshold between the extracted values and the ground-truth. The results demonstrated consistent improvements by LEADS over all baselines. For example, in study characteristics extraction, LEADS achieved 0.68 compared to GPT-4o at 0.55; in arm design, LEADS reached an accuracy of 0.53 while GPT-4o achieved 0.45; in participant statistics, LEADS scored 0.94 compared to GPT-4o's 0.55; in trial results, LEADS obtained 0.78 compared to GPT-4o's 0.45. Open-source LLMs generally performed worse than proprietary LLMs, and medical LLMs underperformed compared to generic LLMs, likely due to their fine-tuning on question-answering datasets. Numeric extraction tasks were found to be more challenging than text extraction tasks. Some target numeric values are not explicitly stated in the raw content but require calculation, such as determining the average age of participants within a defined cohort. Specifically, in participant statistics extraction, LEADS achieved an accuracy of 0.33 while GPT-4o scored 0.20. This difficulty could be partially attributed to the misinterpretation of numerical units and discrepancies in the automatic evaluation process, requiring further human assessment.

We selected a subset and recruited two annotators to manually verify the extraction results (Fig. 4b). We found that LEADS demonstrated a margin of improvement over the baselines, with gains ranging from 1.0 to 55.9. For instance, in the study characteristic extraction task, LEADS achieved an accuracy of 66.2, compared to 59.7 for GPT-4o and 47.8 for the Mistral model. Furthermore, the accuracy of number extraction tasks improved substantially after human

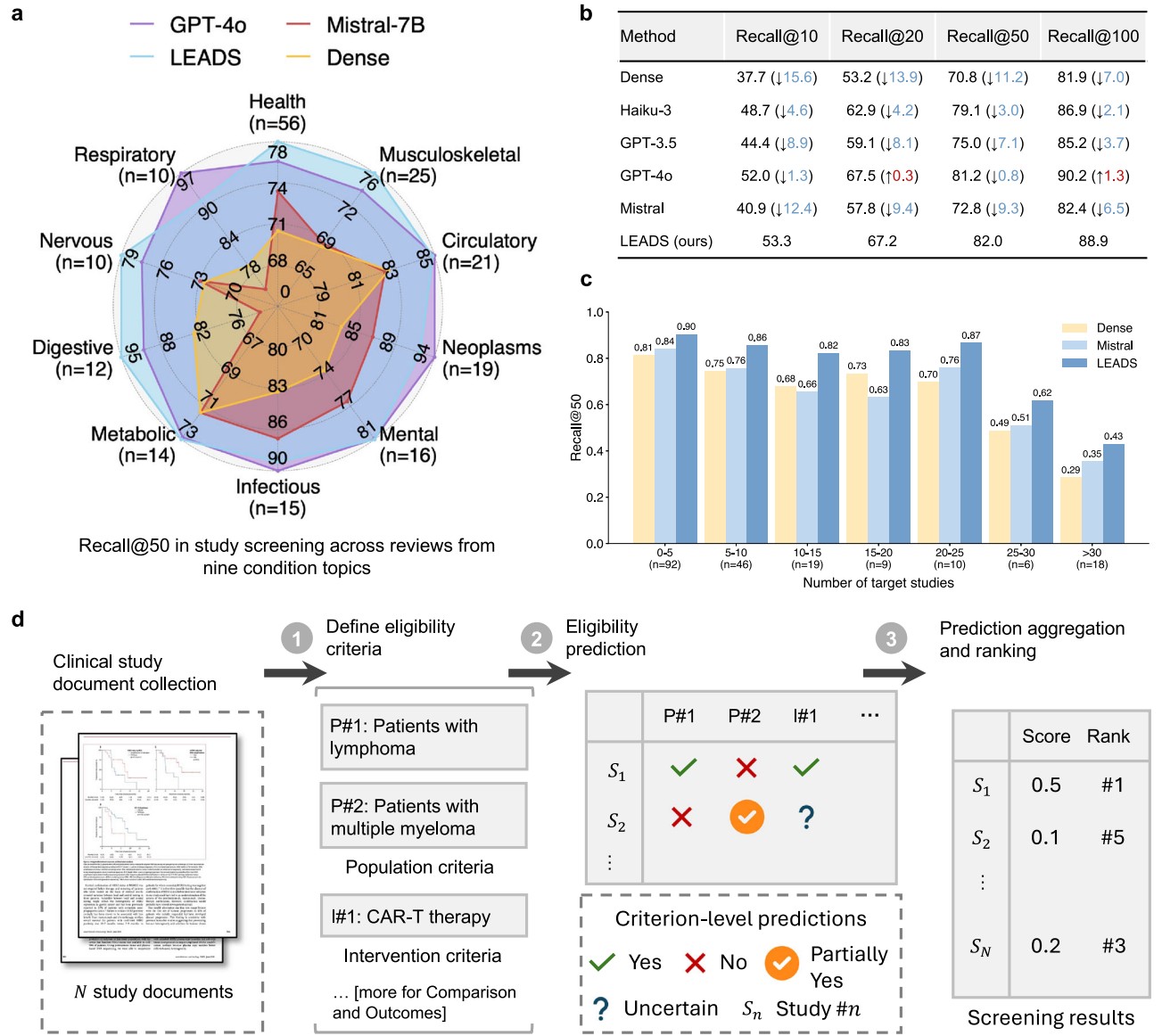

**Fig. 3 | LEADS performs citation screening tasks. a** Radar plot of Recall@50, comparing LEADS to cutting-edge LLMs and dense retrieval across various review condition topics. **b** Recall performance of LEADS comparing to other LLMs and dense retrieval. The information in parentheses indicates the performance change of baselines compared to LEADS. **c** Performance of LEADS and baselines regarding the varied number of target studies for each review. **d** Illustration of how LEADS receives the study inclusion and exclusion criteria defined for target PICO elements, makes eligibility prediction, and ranks the target studies.

annotator calibration. For trial result extraction, LEADS achieved 56.7 in accuracy, outperforming GPT-4o (55.7), GPT-3.5 (51.2), Haiku (54.7), and Mistral (53.2). Across all tasks, LEADS consistently outperformed its generic counterpart, the original Mistral model, by a margin exceeding 20 points in most cases, with differences of 18.4, 34.5, 72.3, 36.2, 24.8, and 3.5 across various metrics.

We further investigated the correlation between extraction performance and input document length (Fig. 4c). Study characteristic extraction tasks tend to have the shortest inputs, primarily relying on study abstracts. In contrast, most other tasks involve inputs averaging around 10,000 tokens, equivalent to approximately 15 pages. The results indicate that cutting-edge LLMs generally exhibit minimal sensitivity to input length within their context windows, reflected in Pearson correlations that are close to zero and slightly negative. Notably, LEADS demonstrates a significant positive correlation with input length ($\rho = 0.22$, $P = 1.5 \times 10^{-4}$), suggesting its invariance to document length.

## Expert collaboration for study screening and data extraction

We conducted a pilot user study to evaluate the practical value of LEADS for medical literature mining. Our focus was on the most time-consuming tasks: study screening and data extraction, to validate two key claims: (1) experts collaborating with AI (such as LEADS) can complete these tasks more quickly than through manual efforts alone, and (2) this collaboration does not compromise the quality of the results. To test these claims, we implemented a two-arm design: one involving experts working independently (Expert-only) and the other combining expert efforts with AI assistance (Expert+AI).

Figure 5a illustrates the setup for screening tasks. Each participant was assigned 10 review topics and tasked with selecting 10 citations from a pool of 30 candidates for inclusion in each review. Participants were randomly assigned to either Arm A (Expert-only) or Arm B (Expert +AI) for half of their review topics to ensure a balanced evaluation. In Arm B, participants can refer to LEADS's assessments: all candidate

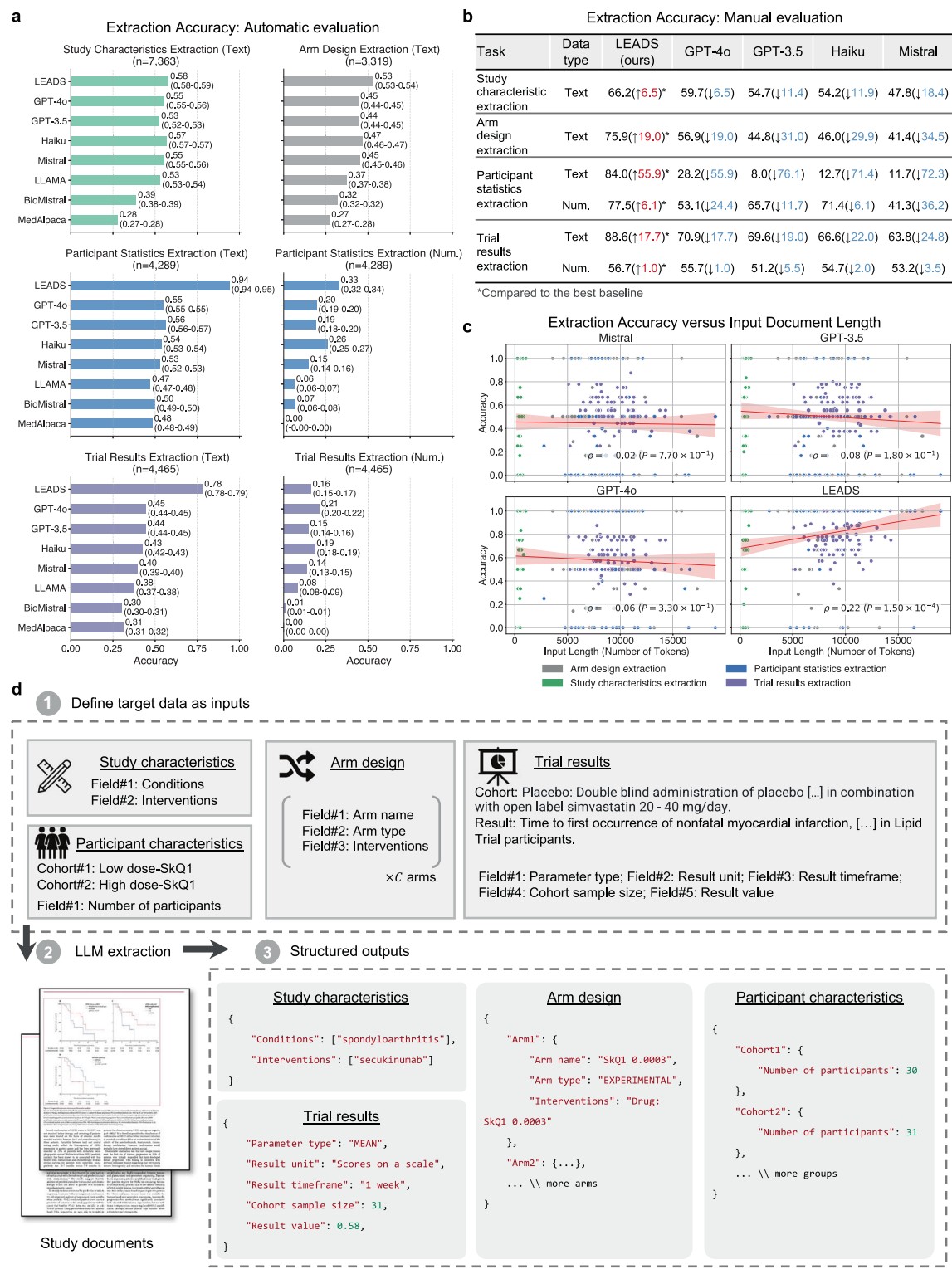

**Fig. 4 | LEADS performs data extraction tasks. a** Mean accuracy of LEADS and other LLMs across four extraction tasks via automatic evaluation. Values in the parentheses indicate the 95% confidence intervals. **b** Accuracy of LEADS and other LLMs across four extraction tasks via manual evaluation. **c** Accuracy of LEADS and other LLMs regarding the varied length of input documents across four extraction tasks via manual evaluation. The red line indicates the regression line for the mean accuracy regarding the input length, and the shaded area is the 95% confidence interval. ρ denotes the Pearson correlation coefficient, and *P* represents the two-sided p-value testing the null hypothesis that there is no correlation between input length and accuracy. **d** Illustration of how LEADS performs four extraction tasks via in-context learning. Based on the definition of the target field and cohorts, LEADS processes study documents and produces structured outputs.

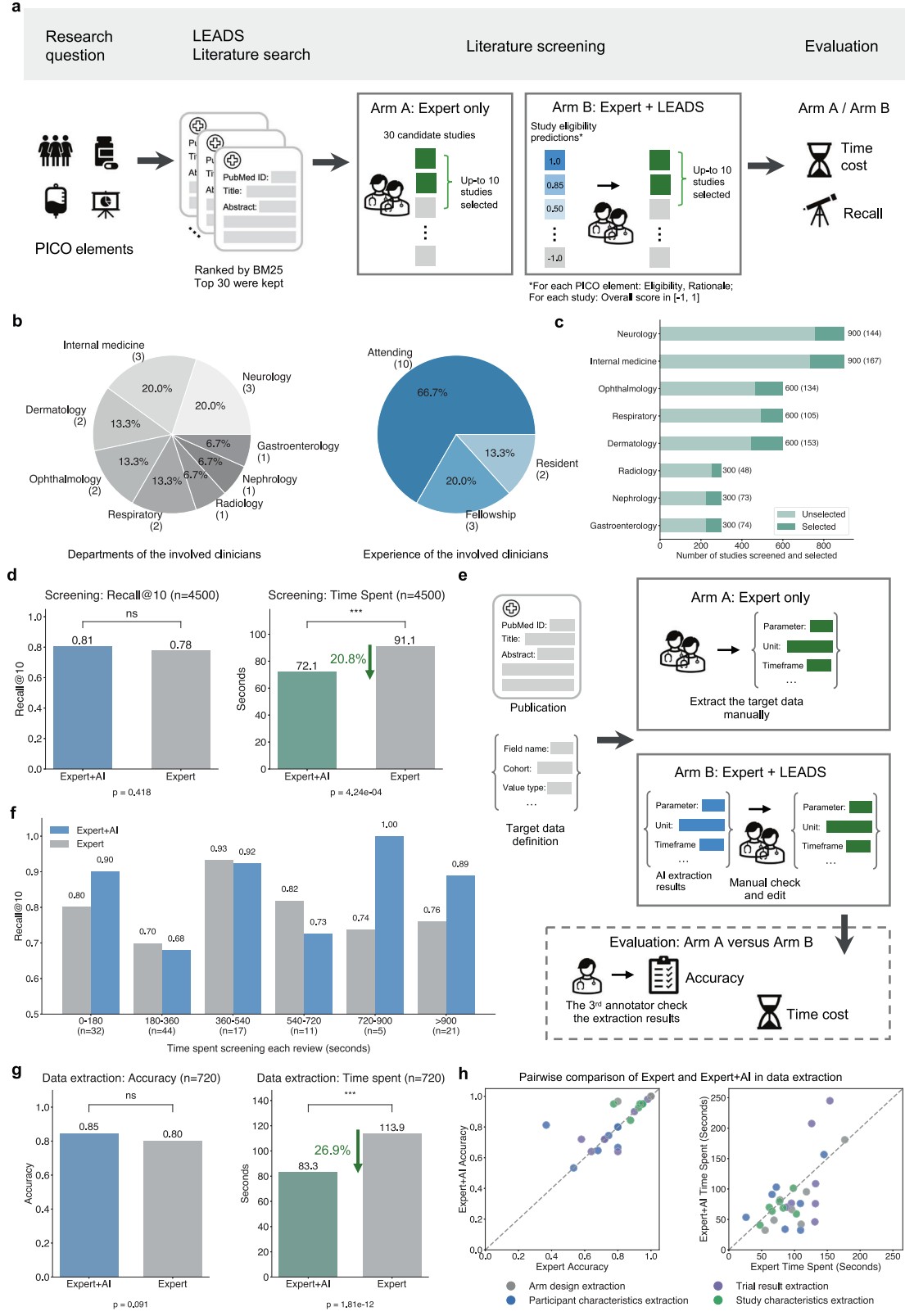

citations were ranked, with additional criterion-level assessments and the explanation provided. This design allowed us to estimate both the time required and the quality of screening results. Extended Fig. 5 includes the forms used in both arms, distributed to participants for completion. We invited 15 clinicians from various departments, such as Neurology, Ophthalmology, and Dermatology, to participate in the study (Fig. 5b). All participants held MD degrees; nine were attending

physicians, while the remaining five were fellows or residents. We ensured that clinicians were assigned review topics aligning with their specialties. Figure 5c shows the number of studies screened and selected by the participants across review topics, with a total of 150 reviews conducted and 4500 studies screened.

Figure 5d presents the average screening quality and time spent. We calculated Recall by comparing the expert-finalized study list for

**Fig. 5 | Pilot user studies for study screening and data extraction. a** The setup of Expert-only arm (Arm A) and Expert+AI arm (Arm B) for study screening tasks. We compare the resulting Recall to evaluate the quality of the time spent per review topic on average to evaluate the speed. **b** The distributions of experts' expertise topics and levels participated in the user study. **c** The number of medical studies screened and selected across review topics by the participants. **d** The overall result quality and time spent per study across two arms in the study screening tasks. The significance levels by the two-sided Mann–Whitney U test of two groups are represented by the *p*-values, in 'ns': non-significance, *: $p < 0.05$, **: $p < 0.01$, and ***: $p < 0.001$. The same applies to other figures if they are not specified specifically.

**e** The setup of Expert-only arm (Arm A) and Expert+AI arm (Arm B) for data extraction tasks. We compare the resulting accuracy to evaluate the quality and the time spent per extraction task on average to evaluate the speed. **f** The screening quality varied across the groups and stratified the time spent working on each individual review topic. Expert+AI yields consistently better performance than Expert only, especially when the tasks are difficult, taking more time to perform screening. **g** The overall accuracy and time spent across two arms in the data extraction tasks. **h** Comparing the accuracy and time spent for data extraction tasks between arms. Each point indicates the average accuracy or time spent on extraction tasks belonging to the same topic.

each review topic against the studies included in the corresponding systematic reviews. Additionally, the time spent on each candidate study was recorded. The results demonstrate that LEADS's support significantly enhances the study screening process. The Expert+AI arm achieved a Recall of 0.81, compared to 0.78 in the Expert-only arm, while reducing the average time spent from 580 s to 449 s, representing a 20.8% relative time savings ($P < 0.001$). Participants noted that AI screening results were particularly helpful for quickly excluding studies with a score of −1, deemed irrelevant, and safely including those scored as 1, which is verified by the distribution of all experts-made decisions and the confusion matrices (Extended Fig. 4). While intermediate-ranked studies still required closer review, the rationale provided by LEADS for PICO eligibility scores served as a valuable aid. Our findings also suggest that these efficiency gains would scale with larger candidate sets. In Extended Fig. 4, we show that nearly all studies excluded by LEADS were also excluded by human reviewers, reinforcing its reliability. Additionally, LEADS achieves Recall@100 over 90%, meaning that in practice, experts could confidently focus primarily on the top 100 results without missing relevant studies.

Figure 5f categorizes completed review topics based on the time spent (e.g., 0–180 s, 180–360 s, etc.), with longer durations generally indicating more challenging reviews. Overall, the Expert+AI arm performed comparably to the Expert-only arm in the less challenging categories, where review times were under 720 s. For example, in the least challenging group (0–180 s), the Expert+AI arm achieved a Recall of 0.9, compared to 0.8 for the Expert-only arm. However, a notable performance gap emerged as the review tasks became more challenging. In the 720–900 s group, the Expert+AI arm achieved a Recall of 1.0, compared to 0.74 for the Expert-only arm. Similarly, in the > 900 s group, the Expert+AI arm achieved a Recall of 0.89, while the Expert-only arm achieved 0.76.

Figure 5e illustrates the setup for the pilot user study on data extraction tasks. Each participant was assigned 90 clinical trial publications and tasked with completing four types of data extraction: study characteristics, arm design, participant characteristics, and trial results, resulting in a total of 360 extraction tasks per participant. Two medical researchers were randomly assigned to Arm A (Expert-only) for half of their extraction tasks and Arm B (Expert+AI) for the other half. In Arm B, participants received LEADS's extraction outputs as references for the target fields. Additionally, participants recorded and reported the time spent on each extraction task. The extraction results were reviewed by two additional annotators and compared against the ground truth to calculate accuracy. The forms used for completing the data extraction tasks are provided in Extended Fig. 6.

Figure 5g presents the average data extraction accuracy and time spent in the two study arms. The Expert+AI arm achieved an accuracy of 0.85, compared to 0.80 in the Expert-only arm, while reducing the average time spent per task from 113.9 s to 83.3 s, resulting in a 26.9% relative time savings. Participants noted that while LEADS's extraction results were not flawless and required verification, they were helpful for quickly locating relevant information within the paper for review and correction. In contrast, participants in the Expert-only arm spent

much of their time thoroughly reading the entire paper, leading to significantly longer task durations.

Figure 5h provides a breakdown across extraction tasks and review topics. We compared accuracy and time between the two arms by aggregating results for studies focused on the same disease areas. The points on the diagonal line indicate that the two arms performed equivalently. The analysis revealed that both arms achieved comparable extraction accuracy overall. Among the tasks, study characteristic extraction showed the highest accuracy, while participant characteristic extraction had the lowest. Regarding time, the Expert-only arm consistently required significantly more time than the Expert+AI arm. The smallest time difference was observed in study characteristic extraction, whereas the difference was much larger for trial result extraction. Participants noted that study characteristics are often found in the paper's abstract, making them easier to extract. In contrast, participant characteristics and trial results are typically located in the main content, making them more challenging and time-consuming to extract.

## Discussion

Performing systematic literature reviews is a cornerstone of evidence-based medicine. However, the process has become increasingly time-intensive and challenging due to the ever-growing literature volume. To address these challenges, large language models (LLMs) have been employed for various literature review tasks[12–14,16,20,30,32,52]. However, existing models have been developed or evaluated on datasets with limited scope, usually covering only tens of systematic reviews and hundreds of studies[39,53]. To overcome these limitations, we created a comprehensive dataset comprising 21,335 systematic reviews, 453,625 publications, and 27,015 clinical trial registries. This dataset establishes a robust foundation for evaluating AI algorithms across a broad spectrum of therapeutic areas. In addition, the data generation pipeline we created also provides a valuable resource for future literature-based scientific discovery.

From the collected literature data, we developed LEADSInstruct, comprising 633,759 instruction data points. Leveraging LEADSInstruct, we fine-tuned LLMs to create LEADS, a foundation model designed for study search, screening, and data extraction, with broad applicability across broad therapeutic areas. The instruction-following capability of LEADS allows it to adapt easily to various input requirements, such as inclusion and exclusion criteria for study screening. Its superior performance was demonstrated through extensive evaluations using the largest benchmark dataset available for medical literature mining. Compared to generic LLMs, many of which are significantly larger, LEADS consistently outperformed them across six validation datasets. It highlights that, when trained on high-quality, curated data with a tailored training process, smaller models can surpass much larger generic models in domain-specific tasks. LEADS exemplifies this by achieving strong performance despite its smaller size, showing the effectiveness of purpose-built models for AI-driven literature search and synthesis.

LEADS provides valuable assistance to medical experts and systematic reviewers by streamlining the literature mining process and

maintaining higher quality than purely manual efforts. In a pilot user study involving 15 clinicians, experts could more effectively identify relevant studies by leveraging LEADS's overall eligibility scores, PICO eligibility predictions, and rationales. This collaboration resulted in an average time savings of 20.8%, a Recall improvement of 5.2%, and a notable 26.1% Recall increase for more challenging review topics. Additionally, the pilot study with two medical researchers demonstrated that LEADS significantly enhances data extraction efficiency and accuracy. By referencing LEADS's extraction results, participants achieved a 6.2% accuracy improvement and reduced their time spent by 26.9%. With its design, LEADS can be seamlessly integrated into the existing TrialMind web platform[32,33] as a backend component, enabling medical professionals to utilize it without any technical barriers.

This study has several limitations. First, while LEADS demonstrates state-of-the-art performance in medical literature mining tasks, its effectiveness relies on the quality of the training data sourced from medical literature and the instruction data generation pipeline. Addressing issues such as potential biases, outdated information, and errors in the data remains a critical area for improvement. Second, the pilot user study setup could be refined to improve the dissemination, e.g., increasing the number of participants and evaluating LEADS in scenarios that more closely simulate real-world tasks, such as screening thousands of candidate citations instead of the 30 used in this study. Third, further research is necessary to optimize LLMs' outputs to integrate AI assistance into systematic review workflows and enhance their practical utility. For example, additional instruction data development is required to cover all tasks necessary for completing systematic literature reviews, such as assessing study quality and evidence uncertainty. Fourth, LEADS is a 7-billion-parameter LLM and requires 20 GB plus GPU memory to deploy locally, which may not be accessible to users who do not own powerful GPUs locally. Finally, despite its promising performance, applying LEADS in medical literature mining must be cautiously approached. Rigorous expert oversight is essential to ensure accuracy and to prevent biased or erroneous outputs. Such validation is particularly critical when using AI for systematic reviews, as errors could lead to the dissemination of misleading or incorrect clinical evidence.

LEADS demonstrates superior performance in literature search, screening, and data extraction, outperforming generic LLMs. It generalizes across a wide range of therapeutic areas without requiring additional training. LEADS showcases its value as an assistant for medical researchers, clinicians, and systematic reviewers by streamlining the literature mining process and facilitating evidence-based medicine. We anticipate that the continued development and validation of foundation models for literature mining will ultimately foster more effective human-AI collaboration to advance healthcare and drug development.

## Methods
### Data collection
The systematic review, publication, and clinical trial data were sourced from publicly available datasets. We began by obtaining a list of medical systematic reviews from the MS2 multi-document summarization dataset[54], which links each review to the studies included in its analysis. This dataset provided an ideal foundation for generating instruction data related to literature search and screening. For PubMed citations, we utilized the PubMed API to retrieve metadata and abstract information[55]. We also attempted to link PubMed citations to clinical trial records on ClinicalTrials.gov, leveraging explicit NCT IDs (clinical trial identifiers) available in some PubMed citations. This linkage enabled the creation of a connection between systematic reviews and trial records, forming the basis for publication search data. To ensure data quality, we removed duplicates, citations lacking essential information, and reviews without associated citations. After processing, the

dataset comprised 21,335 systematic reviews linked to 453,625 publication citations and 8485 systematic reviews linked to 27,015 trial citations, with publication-based reviews averaging 21.26 citations each. To standardize the input for the search query generation and the study eligibility assessment tasks, we adopted GPT-4o to extract the PICO elements from the reviews' abstracts.

The data extraction tasks were built on the links between PubMed citations and clinical trial records. We began by searching the PubMed database and filtering for entries with an associated NCT ID, which indicates a corresponding clinical trial, and full-text availability through PubMed Central (PMC). These criteria ensured that the complete content of each study could be automatically retrieved. For clinical trials, we further filtered for records with reported results to ensure the availability of outcome data. This process resulted in a dataset of 8674 paired publications and clinical trial records. For each trial, we retrieved structured data using the ClinicalTrials.gov API[56], establishing a link between the publication content and structured information on study design, population statistics, and outcome data.

### LEADSInstruct: task formulation
We formulated the key literature mining tasks into instruction data that is suitable for LLM processing. The first task, literature search, according to PRISMA guidelines[43], refers to identifying initial publication records or clinical trial registries from databases. Practitioners typically provide keywords as search queries to search engines, applying basic filters such as year range, publication type, and more, to generate a broad pool of potential study candidates. We define this task as a *search query generation* process, where LLMs take user-defined research questions as input and synthesize or expand the associated key terms for treatments and conditions (Fig. 2a). Users can then review and iteratively refine these terms when retrieving records from search engines.

The second task, citation screening, assesses the eligibility of initially retrieved records based on predefined review protocols, such as PICO elements, to produce a shortlist of studies for review. We define this task as a *study eligibility assessment* process. Unlike previous approaches that rely on LLMs to make overall assessments on whether to include or exclude each citation[37], our approach provides assessment at the criterion level for each specific inclusion and exclusion criterion. This granular approach offers greater flexibility, allowing users to adjust the filtering and ranking of citations by manipulating the criterion-level predictions and the referenced rationales. For example, users can introduce new criteria, select studies based on a subset of criteria, or convert criterion-level predictions into relevance scores. These scores can then be aggregated to rank study eligibility, providing a dynamic and customizable method for prioritizing citations (Fig. 3d).

The third task, data extraction, is a critical step where users review the selected studies to extract key information such as study design and outcomes, enabling the creation of a structured summary for further analysis. We define four subtasks within this process: *study characteristic extraction*, *arm design extraction*, *participant statistics extraction*, and *trial result extraction*, illustrated in Fig. 4d. Study characteristic extraction identifies and extracts predefined fields, such as conditions and interventions, from the study content. Arm design extraction focuses on extracting details about study arms, including their names and types, based on specified fields. Participant statistics extraction requires defining cohorts, typically by treatment groups or observed conditions, and extracting relevant statistics, such as participant counts for each cohort. In trial result extraction, the inputs include the definitions of target cohorts and outcomes, along with the fields of interest, to retrieve information such as parameter type, result unit, timeframe, cohort sample size, and result values.

To optimize AI models for these tasks, we need to develop an instruction dataset consisting of paired input requests and their

expected outputs. Such datasets enable instruction tuning of generic large language models, such as Llama[48] and Mistral[44], to enhance their task-specific performance[57]. Although previous works have created datasets for related tasks[32,39], these datasets are either not aligned with our specific tasks or are too small to train LLMs. In medical applications, adapting LLMs typically requires high-quality instruction data at scales ranging from tens of thousands to millions[49,50,58,59]. The instruction data follows a standardized structure comprising three components: (1) instruction, which describes the task, such as generating search terms; (2) input, which provides the task-specific input, such as the PICO elements defined in a target review; and (3) output, which specifies the expected results that the LLMs should produce.

However, manually creating such datasets is prohibitively labor-intensive, especially as it requires annotators with advanced medical expertise. To address this challenge, it has become common practice to leverage advanced generalist LLMs, such as GPT-4[23], to synthesize outputs based on input instructions, a method known as self-instruct[60]. This approach has been widely adopted in recent developments of medical LLMs[61,62]. However, while GPT-4 can produce high-quality outputs, it is not immune to errors, which can limit the reliability of models trained on synthetic instruction data[63]. To mitigate this issue, we developed a hybrid approach that combines mining instruction data directly from publications and clinical trial registries with augmenting outputs using generalist AI. In total, we compiled 633,759 instruction data across various medical literature mining tasks.

## LEADSInstruct: search query generation

For search query generation tasks, the input is the research question defined using the PICO framework in the review, to generate a set of queries capable of retrieving all ground-truth studies from the literature (Fig. 2a). In our previous study, we observed that queries directly synthesized by GPT-4o typically retrieved fewer than 10% of ground-truth studies in the search results[32]. To improve the quality of search queries generated by GPT-4o, we developed an advanced pipeline that synthesizes query terms from each included study, incorporating an iterative refinement and filtering process to optimize the initial terms. This approach significantly increased the coverage of synthetic queries, retrieving over 80% of ground-truth studies, making it possible for LEADS to surpass GPT-4o while learning from GPT-4o's outputs. This process yields a total of 29,693 samples.

Specifically, consider that there are $N$ ground-truth studies for a review topic. Our aim is to build a comprehensive set of keyword sets for population $\{P_1, ..., P_N\}$ and intervention $\{I_1, ..., I_N\}$, to build the search query that can identify all the ground-truth studies from the literature. As such, for the $n$-th study, we prompted GPT-4o to extract $P_n = \{p_1^n, ..., p_M^n\}$ and $I_n = \{i_1^n, ..., i_M^n\}$ from the study content (The used prompt is in Extended Fig. 21). The number of terms $M$ is set to be at most ten, so the exact extracted numbers are varied. Then, we merge all terms in $P_n$ with the AND logic and merge all $P$ taking the OR logic, leading to the aggregated population-related search query:

$$\mathbf{S}_P = S_P^1 \ \text{OR} \ S_P^2 \ ... \ \text{OR} \ S_P^N, \tag{1}$$

where each $S_P^n = p_1^n \ \text{AND} \ p_2^n \ ... \ \text{AND} \ p_M^n$. Similarly, we obtained the aggregated intervention-related search query:

$$\mathbf{S}_I = S_I^1 \ \text{OR} \ S_I^2 \ ... \ \text{OR} \ S_I^N, \tag{2}$$

where each $S_I^n = i_1^n \ \text{AND} \ i_2^n \ ... \ \text{AND} \ i_M^n$. We hence built the final search query by merging $\mathbf{S}_P$ and $\mathbf{S}_I$, yielding the synthetic target search query $\mathbf{S} = \mathbf{S}_P \ \text{AND} \ \mathbf{S}_I$.

In practice, we validate the generated search query $\mathbf{S}$ by executing it via the PubMed API and calculating the search recall. Queries with a recall below 0.2 are filtered out as poorly generated. The remaining queries are then designated as synthetic ground-truth queries,

achieving an average recall of 0.82. Finally, we wrap this query with a search query generation prompt (Extended Fig. 12) to create an instruction dataset, resulting in 10,262 entries.

To benchmark the search query generation performance for LEADS and the other LLMs, we prompted them to generate the search query $\hat{\mathbf{S}}$ taking the review's PICO definition as the input (Fig. 2a). In addition, we introduced LEADS+ensemble, an extension of LEADS that samples ten queries $\{\hat{\mathbf{S}}_1 ... \hat{\mathbf{S}}_{10}\}$ from the model. Then, we executed all queries and returned the aggregated and deduplicated search results as the final outputs.

## LEADSInstruct: study eligibility assessment

Study eligibility assessment evaluates whether a study meets predefined eligibility criteria based on specific guidelines, structured around PICO elements: Population, Intervention, Comparison, and Outcome. We extracted the study selection criteria defined in systematic reviews and categorized each criterion into P, I, C, or O elements. Assuming that all ground-truth studies included in the reviews meet these criteria, we utilized GPT-4o to generate rationales for criterion-level eligibility assessments (Fig. 3d). Additionally, we included citations retrieved during the search process that were not incorporated into the reviews and used GPT-4o to generate both eligibility predictions and corresponding rationales, creating a balanced instruction dataset. This process resulted in a comprehensive dataset comprising 461,585 review-to-citation eligibility prediction pairs.

In detail, candidate publications or trials were constructed from the search results described earlier. Citations were initially added to the candidate pool, and if fewer than 2000 entries were available, the remaining slots were filled with additional search results generated using other PICO elements. A time constraint was applied to ensure that the studies to be screened were published before the target review paper. For each systematic review, a method must evaluate up to 2000 studies, scoring and ranking them to ensure that ground-truth studies appear at the top. The dataset was split into training, development, and test sets using a 6:2:2 ratio, resulting in 12,801 training reviews, 4217 development reviews, and 4217 test reviews for publications. Due to the high computational cost of requiring LLMs to review 2000 studies per systematic review, a subset of 200 test entries was created for LLM evaluation.

We constructed instruction tuning data for study eligibility assessment based on the training split for publication eligibility prediction. For systematic review purposes, each study must be scored and ranked accordingly. However, the ground-truth data only indicate whether a candidate study is eligible for inclusion in a review. To address this, we prompted GPT-4o to analyze eligibility as shown in Extended Fig. 22. We provided GPT-4 with each study's PICO elements as criteria, the study content, and an overall indication of eligibility. GPT-4 then generated an eligibility analysis, offering a rationale for each criterion. Each criterion was rated as "YES", "PARTIAL", "UNCERTAIN", or "NO", corresponding to scores of 1, 0.5, 0, and −1, respectively. The final eligibility score was calculated as the average of all criterion scores:

$$\text{Final eligibility score} = \frac{\sum(\text{Criterion scores})}{N} \tag{3}$$

These generated analyses were stored as outputs for the instruction data. We then wrapped the analysis, paper content, and criteria with the prompt in Extended Fig. 13 as instruction data. This process was applied across 12,801 reviews and their 2000 candidate studies in each review. Finally, we filtered out studies marked as eligible but with a negative overall score, resulting in a total of 461,585 entries.

## LEADSInstruct: data extraction

The linkage between publications and clinical trial registries facilitates the automated creation of extraction data. On ClinicalTrials.gov, trial records are input by principal investigators, including high-quality structured information such as conditions, interventions, enrollment numbers, study types, and, in some cases, reported results. We assume that clinical trial-related publications linked to these trial records also contain descriptions of this information within their content. Leveraging this connection, we identified 8674 linked publications with full-text availability and corresponding clinical trial records with reported results. By extracting structured trial information and parsing the PDF content of the publications, we generated 58,593 instruction data points for study characteristic extraction, 42,794 for trial result extraction, 34,138 for participant statistics extraction, and 26,387 for arm design extraction (Fig. 1a).

The arm design extraction dataset is constructed from matched publications and their associated trials. Each entry includes the full text and table content of the publication as input, from which arm design details are systematically extracted. The results field serves as the output, containing a list of intervention arms, where each arm specifies a unique label, type (e.g., "EXPERIMENTAL"), description, and the intervention names involved. The ground-truth result fields are extracted from trial reports. This structure enables efficient extraction of arm design information from publication content, with each entry capturing detailed intervention characteristics (Fig. 4d).

The participant statistics extraction dataset is derived from matched publications and their corresponding clinical trials. For each entry, we include the full text of the publication, including any table content, and extract key attributes from the clinical trial reports, such as measure definition, parameter type, unit of measurement, and participant group definitions. Each participant group entry includes a unique group ID, unit, value, and definition. Additionally, the dataset contains a list of results, with each result specifying a group ID, a value, and any relevant notes. Here, results serve as the output, while the other fields constitute the input. The value represents specific participant statistics as defined by the input parameters (Fig. 4d).

The trial result extraction dataset is built from paired publications and their corresponding clinical trials. For each entry, we extract the full text of the publication, including the main content and any text from the tables. From the trial report, we extract the outcome definition, group definition, parameter type, unit of measurement, specified timeframe, denominator unit, and denominator value. Each entry also includes a list of results, each providing a specific value and a descriptive title in a trial report. The input consists of outcome and group definitions, while the other fields are outputs. The value represents a specific outcome of the trial as defined by the input parameters (Fig. 4d).

We construct data extraction instruction tuning data from the training split across four data extraction tasks. For each entry, we format the data with prompts specific to each task: study characteristic extraction (Extended Fig. 14), arm design extraction (Extended Fig. 15), participant statistics extraction (Extended Fig. 16), and trial result extraction (Extended Fig. 17).

## LEADS: model training

All experiments were run in Python 3.12. The detailed software versions are vLLM v0.6.4, post1, openai v1.55.1, transformers v4.46.3, and PyTorch v2.5.1. LEADS was built upon the Mistral-7B-Instruct-v0.3 model[44], chosen for its inherently long context window. We fine-tuned this base model on the LEADSInstruct dataset, resulting in the LEADS model. The training used the data type bfloat16 for enhanced computational efficiency. We trained our model using the AdamW optimizer[64] for one epoch with a batch size of 5. A cosine learning rate scheduler was adopted, with a peak learning rate of $1 \times 10^{-6}$ and a

warm-up phase covering 10% of the training steps. The maximum sequence length was set to 30,000 tokens to accommodate the lengthy full-text literature in data extraction tasks. We implemented the code using PyTorch[65] and the Hugging Face Transformers library[66]. To improve training speed and optimize memory usage, we integrated DeepSpeed ZeRO-3[67] and FlashAttention-2[68] strategies. After completing the instruction tuning process, we obtained the final LEADS model. The instruction tuning was performed on 5 Nvidia A100 80G GPUs over approximately 2.5 days.

## Details of automatic evaluation

Our model selection was designed to encompass a range of general-purpose LLMs with varying capabilities while maintaining a focus on evaluating LEADS against relevant baselines. GPT-4o was chosen as the strongest available model at the time of LEADS' development, serving as a robust upper-bound baseline. GPT-3.5 and Haiku-3 were included as lightweight, less powerful models to assess performance at a lower computational scale. Mistral, as a general-purpose model with a similar architecture to LEADS, was selected as a critical baseline for evaluating the benefits of domain-specific fine-tuning.

For adaptation, we applied in-context learning (ICL) and few-shot prompting for study search but not for study screening or data extraction, due to fundamental differences in how these tasks are approached. Study search typically focuses on a single research question at a time, making refined prompting strategies more practical. In contrast, study screening requires evaluating large volumes of studies in parallel, each with unique characteristics, making it challenging to design a universal ICL setup that works across all cases. A similar limitation applies to data extraction, where the diversity of extracted information further complicates effective prompting strategies.

Study search experiments were conducted for both publication and trial search tasks. We compared LEADS against generative language models by using them to generate queries, which were then executed through the PubMed API or CTGov API to retrieve search results. The selected competing LLMs included GPT-3.5-turbo, GPT-4o, and Claude-3.5-Haiku. Four types of prompts were designed to adapt these models for query generation: zero-shot, few-shot, in-context learning (ICL), and a combination of ICL and few-shot, as illustrated in Extended Fig. 18. Additionally, we evaluated Mistral-7B-Instruct-v0.3, the base model used for training LEADS, as a baseline. Search performance was measured using recall@3000, which is defined as the proportion of ground-truth studies retrieved within the top 3000 search results. This metric was calculated for both publication and trial search tasks. For Mistral and LEADS, the prompts used to generate search queries are shown in Extended Fig. 12. To further enhance search performance, we implemented an ensemble approach (LEADS + ensemble). This approach combines all possible sets of keywords generated by LEADS for population and intervention terms, maximizing coverage and retrieving the most comprehensive results possible.

In the study screening experiments, each systematic review involves 2000 candidate citations that need to be scored and ranked based on PICO elements. The performance of the selected methods is evaluated using recall@$K$, where $K$ is set to 10, 20, or other specified values. We tested two types of methods: dense retrieval models and LLMs. Dense retrieval models generate text embeddings for the PICO elements and the content of candidate studies, calculate cosine similarity scores between these embeddings, and rank the studies accordingly. While these models are computationally efficient, they generally exhibit lower performance. For dense retrieval, we used OpenAI's text-embedding-small model. For LLM-based study screening, we tested GPT-3.5-turbo, GPT-4o, and Claude-3.5-Haiku, employing two types of prompts: a simple prompt that assigns a score from 1 to 10 (Extended Fig. 19), and an advanced prompt (Extended Fig. 20) that uses a two-stage approach, first generating

criteria and then scoring based on those criteria. We report the performance of each baseline with their best prompt. For LEADS and its base model, Mistral, we used the same prompt format (Extended Fig. 13) to generate eligibility predictions. The final score of each study was calculated using the same methodology applied during the creation of instruction data.

Data extraction tasks include study characteristics, arm design, participant statistics, and trial result extraction. We employ both proprietary LLMs (GPT-3.5-turbo, GPT-4o, and Claude-3.5-haiku) and open-source LLMs (Meta-Llama-3-8B-Instruct[48], Mistral-7B-Instruct-v0.3[44], MedAlpaca[50], and BioMistral[49]). The first two open-source models are popular LLMs for general domains, while the latter two are fine-tuned LLMs for the medical domain. For all these models and LEADS, we use the same prompts (Extended Figs. 14–17) to generate extraction results. We evaluate extraction performance through both automated testing and manual evaluation. For automated testing, we assess fields containing numerical and textual data. Numerical fields require exact matching, while textual fields use soft matching. For textual fields, we use text embeddings to calculate the similarity between predictions and ground-truth values, applying a cosine similarity threshold of 0.75. Predictions exceeding this threshold are considered correct. Participant statistics extraction and trial result extraction both contain numerical fields, so we report results for text and numerical fields separately. For manual evaluation, we randomly select 75 studies from each data extraction task, totaling 300 cases per model prediction. The selected studies are the same across all models used for evaluation. Each predicted result is manually compared with the ground truth for each field, and we report results separately for both textual and numerical fields.

### Details of pilot user study

To prepare the data for the study screening task, we collected 30 candidate studies for each systematic review, with up to 10 of these studies included in the reference systematic review paper, which we used as the reference answer. Each systematic review was categorized into a therapeutic area. Clinicians were asked to select one area aligned with their expertise, and 10 review topics were assigned within their chosen area. Clinicians were tasked with selecting up to 10 studies per review topic that met the PICO framework. Of the 10 review topics, 5 were completed under the Expert-only arm, where clinicians made decisions independently. In the remaining 5 topics, clinicians participated in the Expert+AI arm, where they received assistance from LEADS.

To prepare the data for the Expert+AI arm, we ran LEADS to assess the PICO criteria from each target review, generating study eligibility assessments for all 30 candidate studies. These predictions included an overall eligibility score, PICO eligibility assessments, and rationales. The results were compiled into a spreadsheet ranking the 30 candidate studies, which served as a reference for clinicians in the Expert+AI arm. Extended Fig. 5 provides examples of the forms used in both arms, distributed to participants for completion. The study involved 15 clinicians from various specialties, including Neurology (3), Ophthalmology (2), Dermatology (2), Internal Medicine (2), Respiratory Medicine (2), Radiology (1), Gastroenterology (1), and Nephrology (1). Among them, nine are attending physicians, three are in fellowship, and two are in residency, ensuring a diverse and representative participant pool for the study.

For the data extraction user study, we collected 90 clinical trial studies covering a range of topics from a variety of specialties, including Ophthalmology, Dermatology, Neurology, Internal Medicine, Radiology, Nephrology, Alzheimer's Disease, Cardiology, and Gastroenterology, with 10 studies per specialty. Each study included four distinct tasks: study characteristic extraction, participant statistics extraction, arm design extraction, and trial result extraction, resulting in a total of 360 tasks. We invited two medical researchers to

participate in the study. Each participant was assigned 180 in the Expert-only arm and another 180 in the Expert+AI arm. For the Expert+AI arm, LEADS was used to perform the four data extraction tasks, generating AI outputs that were provided as references for the participants. The forms for both arms, sent to the participants for completion, are shown in Extended Fig. 6. To evaluate the results, two additional annotators reviewed the submitted outputs, assessing each field as either correct or incorrect. These assessments were then used to compute the overall extraction accuracy for the study.

### Statistics & reproducibility

Since this is not an interventional study, no statistical method was used to predetermine sample size, the experiments were not randomized, and the Investigators were not blinded to allocation during experiments and outcome assessment. The sample size was then determined by data acquisition, followed by a data exclusion and filtering pipeline. We removed duplicates, citations lacking essential information, and reviews without associated citations.

### Reporting summary

Further information on research design is available in the Nature Portfolio Reporting Summary linked to this article.

### Data availability

The PubMed publication data are publicly available at https://pubmed.ncbi.nlm.nih.gov/download/. The PubMed Central publication data are publicly available at https://pmc.ncbi.nlm.nih.gov/tools/ftp/. The clinical trial records are publicly available at https://aact.ctti-clinicaltrials.org/downloads. The LEADSInstruct data generated in this study have been deposited in the Hugging Face database under accession code: https://huggingface.co/datasets/zifeng-ai/LEADSInstruct, including training, testing, and development datasets for search query generation, study eligibility evaluation, study characteristic extraction, trial result extraction, participant statistics extraction, and arm design extraction. Source data are provided with this paper.

### Code availability

The implementations of LEADS can be accessed at https://github.com/RyanWangZf/LEADS. The trained model weights are available at https://huggingface.co/zifeng-ai/leads-mistral-7b-v1.

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

## Acknowledgements

Q.J. and Z.L. were supported in part by the Division of Intramural Research (DIR) of the National Library of Medicine (NLM), National Institutes of Health. B.A. was supported in part by the Intramural Research Programs of the National Institute of Diabetes and Digestive and Kidney Diseases (project number ZIA/DK075149 to B.A). J.S. was partially supported by NSF awards SCH-2205289, SCH-2014438, and IIS-2034479. Y.P. and J.F.R. were supported by the National Library of Medicine [grant numbers R01LM014306, R01LM014306, R01LM014573].

## Author contributions

Z.W. and J.S. conceived the project. Z.W. and L.C. developed the methodology, curated the training data, and led model development and evaluation. J.C., N.W., B.A., H.-J.C., C.-I.C., M.E., M.K.G., S.-H.K., Y. Li, Y. Liu, Y. Luo, H.O., J.F.R., I.S., J.J.W., Z.X., and C.M.Z. contributed to the design and execution of the multi-institutional user study, including study selection and data extraction tasks, and provided critical feedback on system integration into expert workflows. Q.J., K.K., Y.P., and Z.L. coordinated project planning, experimental design, and model validation. All authors discussed the results and contributed to the final manuscript.

## Competing interests

The authors declare no competing interests.
