## [Transparent Peer Review file · Nature Communications]

A foundation model for human-AI collaboration in medical literature mining

Corresponding Author: Professor Jimeng Sun

Version 0:

Reviewer comments:

Reviewer #1

(Remarks to the Author)

TO THE AUTHORS: This paper looks at the creation of a foundation model for systematic review of the scientific literature called LEADS. The authors construct a IFT dataset (LEADSInstruct) curated from systematic reviews, clinical trials, and registries. They assess their model against several generic LLMs on six tasks and demonstrate superior performance across the board, and show that it improves expert workflows by assisting with reference management and ultimately saving time in a prospective study.

KEY RESULTS

- Creation of a benchmark of six literature mining subtasks: optimizing search queries to maximize study identification, evaluating study eligibility, extracting characteristics, extracting arm design, extracting participant statistics, and extracting study results.
- Creation of LEADS – an IFT tuned version of Mistral-7B for these tasks.
- Pseudo-prospective benchmarking showing superior performance to GPT4o on study search performance, and equivalent performance on study screening performance.

Validity

No concerns

Originality and Significance

The control frontier models (GPT-4o, Haiku-3, etc...) are well addressed by including both zero-shot, ICL, few-shot experiments, and a pseudo-prospective study. I'm surprised that the pseudo-prospective study results showing equivalent or superior performance to GPT-4o and Deep Research aren't highlighted more in the text?

Data and Methodology

No concerns.

Appropriate use of statistics

The figures are very polished – nice work.

Conclusions

Reasonable.

Suggested improvements

None

References

Acceptable

Clarity and context

Very well written.

Assessment of my expertise and areas that might be out of scope

I'm an expert on this topic and confident in my review.

(Remarks on code availability)

Acceptable.

Reviewer #2

(Remarks to the Author)

Thank you for the meticulous response to the reviewers' comments. We are satisfied with the revision of the manuscript. One final comment is that running the model on Colab via HuggingFace requires significant compute resources (likely more than 20GB RAM). Therefore, the model is not accessible unless one has a powerful GPU. Please mention this as a limitation.

(Remarks on code availability)

The links to the dataset and for the model in HuggingFace are available, and the GitHub repo has a good description of each. We were able to run the model but at a non-trivial compute cost.

Reviewer #3

(Remarks to the Author)

The authors developed the AI foundation model for evidence synthesis. The model is trained on 633,759 instruction data points in LEADSInstruct, curated from 21,335 systematic reviews, 453,625 clinical trial publications, and 27,015 clinical trial registries. LEADS has relatively high accuracy and recall, and the study is of considerable significance, but there are some issues that need to be addressed:

- The authors stated four cutting-edge generic large language models (LLMs) on six tasks, could you please specify which six tasks you are referring to?
- In the second paragraph of the introduction section, please consider citing the following references: (1) Luo X, Chen F, Zhu D, et al. Potential Roles of Large Language Models in the Production of Systematic Reviews and Meta-Analyses. *J Med Internet Res*. 2024;26:e56780. Published 2024 Jun 25. doi:10.2196/56780; (2) Mahuli, S., Rai, A., Mahuli, A. et al. Application ChatGPT in conducting systematic reviews and meta-analyses. *Br Dent J* 235, 90–92 (2023). <https://doi.org/10.1038/s41415-023-6132-y>
- The three fundamental tasks in systematic review you mentioned are, in my opinion, very important. However, I have significant concerns regarding citation screening. As you know, due to the black-box nature of generative AI, ensuring transparency throughout the screening process is crucial. It greatly affects users' trust in your model. So, does LEADS offer transparency and visibility during this process?
- I'm quite curious why the authors didn't include risk of bias assessment in LEADS, especially since this area is already well-established (eg., doi:10.1001/jamanetworkopen.2024.12687). What was the rationale or criteria for selecting the tasks included in the system?
- The discussion section should include the idea of whether LEADS should be developed into a product, such as a web-based app, to make it more accessible to a wider audience.

(Remarks on code availability)

I'm not from the computer science field, so I can't understand the code.

Version 1:

Reviewer comments:

Reviewer #3

(Remarks to the Author)

I believe the authors have adequately addressed my concerns, and I consider this work suitable for publication. I have no further comments.

(Remarks on code availability)

I am not a computer science expert, and I do not understand code.

Dear Reviewers,

Thank you again for your thoughtful and constructive feedback during the initial review. We appreciate your continued engagement and have carefully addressed the remaining minor comments in this round. Please find our point-by-point responses below.

Cheers,
Jimeng Sun

Response to Reviewer #1:

Point 1

The control frontier models (GPT-4o, Haiku-3, etc...) are well addressed by including both zero-shot, ICL, few-shot experiments, and a pseudo-prospective study. I'm surprised that the pseudo-prospective study results showing equivalent or superior performance to GPT-4o and Deep Research aren't highlighted more in the text?

Response:

Thanks for your suggestion. We agree it is an interesting finding and have also mentioned this pseudo-prospective evaluation result in the main text:

- In the last paragraph of "Introduction"
- In the second paragraph of "Overview of LEADS and LEADSInstruct"
- In the second paragraph of "Synthesizing literature search queries for target studies"
- In the second paragraph of "Automated assessment and ranking of study eligibility"

Response to Reviewer #2:

Point 1

Thank you for the meticulous response to the reviewers' comments. We are satisfied with the revision of the manuscript. One final comment is that running the model on Colab via HuggingFace requires significant compute resources (likely more than 20GB RAM). Therefore, the model is not accessible unless one has a powerful GPU. Please mention this as a limitation.

Response:

Thanks for your suggestion. We have acknowledged that limitation in the Discussion section.

Response to Reviewer #3:

Point 1

The authors stated four cutting-edge generic large language models (LLMs) on six tasks,

could you please specify which six tasks you are referring to?

Response:

Thanks for your notice. We have updated the “Abstract” and “Introduction” to clarify the six tasks: search query generation, study eligibility assessment, study characteristics extraction, participant statistics extraction, arm design extraction, and trial result extraction. They are illustrated in Fig. 1a of the paper.

Point 2

In the second paragraph of the introduction section, please consider citing the following references: (1) Luo X, Chen F, Zhu D, et al. Potential Roles of Large Language Models in the Production of Systematic Reviews and Meta-Analyses. *J Med Internet Res*. 2024;26:e56780. Published 2024 Jun 25. doi:10.2196/56780; (2) Mahuli, S., Rai, A., Mahuli, A. et al. Application ChatGPT in conducting systematic reviews and meta-analyses. *Br Dent J* 235, 90–92 (2023). <https://doi.org/10.1038/s41415-023-6132-y>

Response:

Thanks for directing us to these relevant references. We have included them in the “Introduction” and “Discussion” of the paper.

Point 3

The three fundamental tasks in systematic review you mentioned are, in my opinion, very important. However, I have significant concerns regarding citation screening. As you know, due to the black-box nature of generative AI, ensuring transparency throughout the screening process is crucial. It greatly affects users' trust in your model. So, does LEADS offer transparency and visibility during this process?

Response:

Thank you for raising this important point. We agree that transparency is critical, especially in citation screening tasks where LLMs can act as black boxes. We've noticed that many prior works rely on binary inclusion/exclusion predictions, which limit user understanding and control [1,2,3].

In contrast, LEADS introduces criterion-level predictions with natural language rationales. Rather than a single binary label, each study is evaluated along multiple inclusion criteria, with justifications provided for each. This design supports ranking, filtering, and auditing during expert review, enhancing transparency and flexibility. An example output from LEADS can be found here: https://ryanwangzf.github.io/reviews/Cardiology_16235290.html

[1] Syriani, E., David, I. & Kumar, G. Assessing the ability of chatgpt to screen articles for systematic reviews. arXiv preprint arXiv:2307.06464 (2023)

[2] Sanghera, R., Thirunavukarasu, A. J., El Khoury, M., O'Logbon, J., Chen, Y., Watt, A., ... & Soltan, A. A. (2025). High-performance automated abstract screening with large language model ensembles. *Journal of the American Medical Informatics Association*, 32(5), 893-904.

[3] Trad, F., Yammine, R., Charafeddine, J., Chakhtoura, M., Rahme, M., El-Hajj Fuleihan, G., & Chehab, A. (2025). Streamlining systematic reviews with large language models using prompt engineering and retrieval augmented generation. *BMC Medical Research Methodology*, 25(1), 1-9.

Point 4

I'm quite curious why the authors didn't include risk of bias assessment in LEADS, especially since this area is already well-established (eg., doi:10.1001/jamanetworkopen.2024.12687). What was the rationale or criteria for selecting the tasks included in the system?

Response:

Thanks for your question and notice of a relevant paper [1]. We agree that risk of bias (RoB) is an important aspect of systematic reviews. Our work focuses on search, screening, and data extraction: tasks where large-scale training data can be automatically generated from linked reviews and publications.

In contrast, RoB assessment requires fine-grained, expert-annotated study-to-risk labels, which are limited. For example, the cited work [1] evaluates LLMs on only 30 studies. In our setting, building a fine-tuned LLM would require at least thousands of studies annotated. We acknowledge this limitation in the Discussion and see scalable RoB data construction as an important direction for future work (please refer to the 3rd point in the 4th paragraph of "Discussion").

[1] Lai, H., Ge, L., Sun, M., Pan, B., Huang, J., Hou, L., ... & Estill, J. (2024). Assessing the risk of bias in randomized clinical trials with large language models. *JAMA Network Open*, 7(5), e2412687-e2412687.

Point 5

The discussion section should include the idea of whether LEADS should be developed into a product, such as a web-based app, to make it more accessible to a wider audience.

Response:

Thank you for your comments. Extended Figs. 5 and 6 illustrate how clinicians used LEADS' outputs as a reference for study screening and data extraction. This study focuses on developing a model that generates explainable outputs to support human-AI collaboration in these tasks.

Regarding usability, our previous work [1] introduced a web-based platform designed for clinicians and researchers without coding expertise. The interface follows the PRISMA workflow, covering search, screening, and data extraction stages. A demo video demonstrating the platform's functionality is available at [\[https://www.youtube.com/watch?v=5VKt_s4X5M0&ab_channel=Alforhealth\]](https://www.youtube.com/watch?v=5VKt_s4X5M0&ab_channel=Alforhealth). LEADS can be seamlessly integrated into this web tool for practical use. In the revision, we have highlighted the potential of LEADS to be integrated into the existing software to enable medical professionals to use it.

[1] Wang, Z., Cao, L., Danek, B., Jin, Q., Lu, Z., & Sun, J. (2024). Accelerating clinical evidence synthesis with large language models. *arXiv preprint arXiv:2406.17755*.